# DOES CALIBRATION AFFECT HUMAN ACTIONS?

## ABSTRACT

Calibration has been proposed as a way to enhance the reliability and adoption of machine learning classifiers. We study a particular aspect of this proposal: what is the effect of calibrating a classification model on the decisions made by non-expert humans consuming the model's predictions? We perform a Human-Computer-Interaction (HCI) experiment to ascertain the effect of calibration on (i) trust in the model, and (ii) the correlation between decisions and predictions. We also propose further corrections to the reported calibrated scores based on Kahneman and Tversky's prospect theory from behavioral economics, and study the effect of these corrections on trust and decision-making. We find that calibration is not sufficient on its own—the prospect theory correction is crucial for increasing the correlation between human decisions and the model's predictions. While this increased correlation suggests higher trust in the model, responses to "Do you trust the model more?" are unaffected by the method used.

## 1 INTRODUCTION

Machine learning (ML) models often assume the role of assistants rather than sole decision-makers, aiding humans in making the final decision that leads to a tangible action. However, in some areas where decisions are made, such as weather forecast, medical diagnosis, risk assessment, fraud detection, and financial forecasting, it is essential that the model not only provides a predicted class but also associates its classification with a probability.

As a simple motivating example, consider a couple planning an outdoor wedding. It is not sufficient for a model to predict that there will be no rain on their wedding day; the model must also provide a probability. It is quite likely, that even if this probably is less than 50%, but not negligible (e.g., 30%), the couple will organize their wedding indoors. That is, the couple is not concerned only with whether the model predicts rain or not; they are actually interested in the probability of rain.

Indeed, classification models are typically inherently probabilistic, that is, their raw outcome is a probability (or a set of probabilities) that is translated into a class prediction. It is common practice to present a probability to the user alongside each prediction; this probability is often referred to as a *confidence score*. When confidence scores are reported, the human decision-maker consuming it must use it in a way that is consistent with what the machine learning expert had in mind when designing the model.

The first consideration we can make is if the model is calibrated. That is, does the probability of the predicted class align with the confidence score provided? Many out-of-the-box approaches lead to classifiers that produce over-confident predictions. For instance, a model might predict a rainy day with a 90% probability. However, if we analyze all model predictions calling for rain with a 90% probability, only 70% might correspond to actual rainy days. Specifically, it is well known that modern neural networks suffer from over-confident predictions Guo et al. (2017).

A well-calibrated model should produce a probability that accurately reflects the true likelihood that the event will occur. For example, if a well-calibrated model predicts a rainy day with an 80% probability, it should rain on approximately 80% of the days with such a prediction. A number of papers have proposed methods to produce classifiers that are calibrated ; see Gupta (2023, Chapter 1) for a recent review of the literature. For completeness, we also review some calibration methods in Appendix A.1.

However, despite the existence of many different calibration methods, which compose an entire field in machine learning, little is known about how *humans* react to a calibrated model in comparison to the original uncalibrated model. A human-focused evaluation of calibration is natural to ask for. Such an evaluation offers multiple benefits as noted next. First, human evaluation provides a means to assess the real-world impact and effectiveness of the model. While traditional evaluation metrics offer quantitative insights, human subjects can provide qualitative feedback that captures nuances and contextual considerations that may be missed by automated assessments alone (Bostrom & Yudkowsky, 2018). Second, human evaluation allows for assessing the usability and user experience of the model. Feedback from humans can provide valuable insights into how well the model meets the needs and expectations of users (Kaplan & Haenlein, 2019). Furthermore, human evaluation helps to ensure that the model aligns with ethical and legal standards. It allows for the examination of the fairness, transparency, and accountability of the model, as well as its compliance with regulations and guidelines (Mittelstadt et al., 2016).

## 1.1 An additional layer of Kahneman and Tversky's prospect theory

In this paper, we introduce an additional layer on top of the existing calibration methods, based on the *prospect theory* from behavioral economics (Kahneman & Tversky, 1979; 1992). Prospect theory provides a behavioral or psychological framework that seeks to explain how individuals perceive and evaluate probabilities of potential gains and losses. We posit that considering prospect theory is crucial since it is the perceived probability that influences individuals' decision-making and trust in the system, and not the actual reported probability. Prospect theory uses a weighting function to describe how people subjectively weigh probabilities. In a nutshell, events with very low probabilities (i.e., close to 0) are often perceived as more likely than they truly are, while events with very high probabilities (i.e., near 1) are perceived as less likely than they truly are.

In our approach, we propose transforming each calibrated probability based on the inverse of the prospect theory weighting function to better match user perceptions. Thus, by using the inverse function, we can derive a probability that users perceive as matching the original prediction. For instance, if individuals perceive an event with a reported probability of $90\%$ as actually having an $80\%$ chance of occurring, then for an actual probability of $80\%$, we would report it as $90\%$ to align with their perception.

## 1.2 Experimental method and summary of findings

We performed our study for the problem of rain forecasting. As previously discussed, this domain is commonly associated with probabilistic predictions and is easy for most to understand. We used a neural network as the uncalibrated model. We tested several calibration methods and selected the isotonic regression method, which performed slightly better than other methods.

We then performed a survey with real humans to evaluate our prospect theory-based calibration method. Specifically, we presented each participant with a rain prediction system. Following each prediction, participants were asked to indicate, assuming they are planning an outdoor activity for that day, how likely they are to cancel the activity. Additionally, they were requested to rate their level of trust in the predictive model.

The rain prediction system could be our method (prospect theory correction on top of isotonic regression), or one of four baselines. Namely, the uncalibrated neural network model, the model calibrated with isotonic regression, the prospect theory correction directly over the uncalibrated model, and our method with the reported outcome (rained or not) drawn uniformly from $\{0, 1\}$. (The last baseline corresponds to changing the outcome instead of the reported probability, whereas the other three correspond to changing the reported probability.)

Our results show that there is no apparent difference in the level of trust reported by the participants. Nonetheless, in the correlation between participants' decision to cancel outdoor activities and the models' rain predictions, we observed a significant difference. Our method resulted in a significantly higher correlation in comparison to all other baselines. These results indicate the utility of incorporating prospect theory into calibration methods.

To summarize, the main contribution of this paper is that it provides a novel approach that incorporates the principles of prospect theory into existing calibration methods. Furthermore, it describes

an evaluation survey with human participants to validate the effectiveness of this approach and investigate the general impact of model calibration on humans.

## 2 RELATED WORK ON DECISION-MAKING AND CALIBRATION

The relationship between calibration, AI-assisted decision-making, and trust, appears to be relatively under-explored, but we review several relevant papers. Rechkemmer & Yin (2022) examine in their survey how the following performance indicators affect people's trust in the model: the stated accuracy, the observed accuracy, and the level of confidence score accompanying each prediction of the model. Their findings revealed that the level of confidence had minimal impact on trust, whereas the stated accuracy and observed accuracy of the model had a more substantial influence. In another study conducted by Yu et al. (2019), they examined the circumstances under which people trust artificial intelligence systems and how they perceive the system's predictions when they are required to collaborate with it in making decisions. Their survey results unveiled that for systems with 70% accuracy or higher, people tended to increase their trust in the system after a series of experiments. Conversely, in systems with lower accuracy people's trust is decreasing and they rely more on their own judgments.

Zhang et al. (2020) examined through a survey whether the presentation of a confidence score and local explanations that accompany each prediction of the model, influence the calibration of people's trust in the model. That is, to adjust the level of people's trust in the model so that it more accurately reflects its actual performance. The results show that the level of confidence does help to calibrate trust, but is not sufficient to improve shared decision-making, but also depends on other factors, such as whether the person brings enough unique knowledge to supplement the model's errors. In addition, it is shown that local explanations do not create a noticeable effect on trust calibration. Barbosa et al. (2022) also conducted a study aimed at calibrating people's trust in a machine-learning model. They tested whether adding class probabilities as a confidence score, when classifying animals in images, will help to calibrate people's trust. Their results revealed that incorporating class probabilities, especially in instances where distinguishing between different animals was challenging, did not lead to a significant improvement, and even increased skepticism in some cases.

It is commonly believed that the confidence value should be a well-calibrated estimate of the probability that the predicted label matches the true label (Gneiting et al., 2007). However, Vodrahalli et al. (2022) have suggested that AI models that report confidence values that do not align with the true probabilities can encourage humans to follow AI advice. They demonstrate through a survey with humans that presenting AI models as more confident than they actually are can enhance human-AI performance in terms of accuracy and confidence of the humans in their final decision. They train a model to predict human incorporation of AI advice and use this to transform the AI's prediction confidence. This finding is validated across different tasks and supported by simulation analysis. In our work, we add an additional layer based on the prospect theory over a calibrated model and show an improvement in people's decision-making. Furthermore, we focus on situations in which the model must predict a probability rather than predicting a class and a confidence value. Finally, we test several baselines including an uncalibrated model, a calibrated model, the uncalibrated model with the prospect theory, and our method with random outcomes.

## 3 THE PROSPECT THEORY CORRECTION

According to prospect theory, people evaluate probabilities based on a reference point, typically the status quo or their current situation. Their evaluations are influenced by whether an outcome is perceived as a gain or a loss relative to the reference point. People often exhibit risk-seeking behavior when facing losses and risk-averse behavior when facing gains, as they strive to avoid losses and maintain the reference point. This phenomenon is known as *loss aversion* (Kahneman & Tversky, 1979; 1992).

Incorporating prospect theory into model calibration can improve the accuracy and realism of the model's predictions, making it more effective in capturing human behavior. Since prospect theory accounts for the biases and heuristics that individuals rely on when making decisions, incorporating these biases can better simulate and predict how people respond to different scenarios.

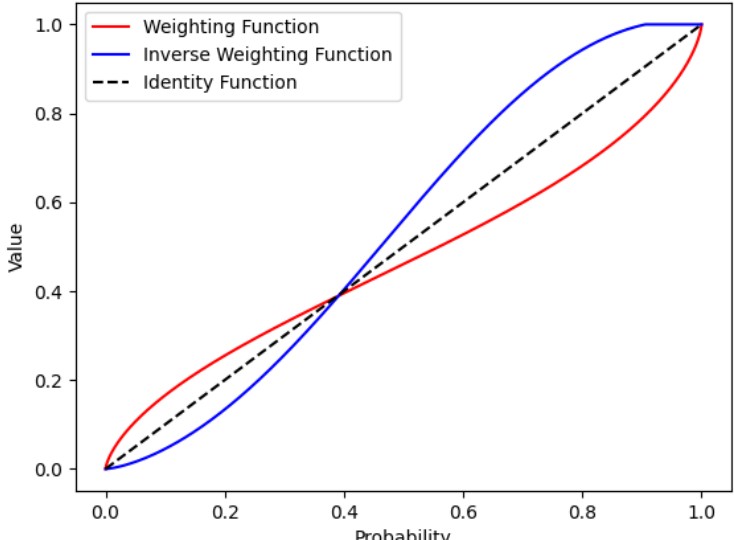

Figure 1: Prospect theory uses a weighting function to transform reported probabilities into perceived probabilities. The red curve represents the weighting function (equation 1) that maps reported probabilities on the X-axis to perceived probabilities on the Y-axis; the blue curve is its inverse (equation 2). Plots are with $\gamma = 0.71$.

Prospect theory uses a non-linear weighting function to describe how people subjectively weight probabilities. The standard weighting function is

$$w(p) = \frac{p^\gamma}{(p^\gamma + (1-p)^\gamma)^{\frac{1}{\gamma}}}, \tag{1}$$

with the parameter $\gamma \in (0, 1]$ describing the amount of over- and underweighting. The weighting function has different $\gamma$ values for gains and for losses (Kahneman & Tversky, 1992). To guarantee a monotone probability weighting function, the value of $\gamma$ has to be larger than $0.279$ (Rieger & Wang, 2006; Ingersoll, 2008; De Giorgi & Legg, 2012). This function captures the diminishing sensitivity to gains and increasing sensitivity to losses.

In our approach, we propose to transform each calibrated probability using an approximation of the inverse of the weighting function:

$$w^{-1}(p) \approx \frac{p^{\frac{1}{\gamma}}}{(p^{\frac{1}{\gamma}} + (1-p)^{\frac{1}{\gamma}})^{\frac{1}{\gamma}}}. \tag{2}$$

To illustrate, we have plotted in Figure 1 the weighting function (equation 1) and its inverse (equation 2) for $\gamma = 0.71$.

## 4 EXPERIMENTAL DESIGN

### 4.1 DATA AND CALIBRATION MODEL

We now describe our dataset and model, along with the calibration method used and details of the survey with human subjects.

We use a rain forecasting dataset. We chose this dataset because it offers predictions that are easily comprehensible, making it accessible even to non-experts.[1] This particular dataset spans approximately a decade of daily weather observations from numerous Australian weather stations. Australia, being a country where rainfall occurs for nearly half of the year, presents an ideal setting

---

[1]https://www.kaggle.com/datasets/jsphyg/weather-dataset-rattle-package

| Model | Acc↑ | F1↑ | ECE↓ | NLL↓ | Brier↓ |
|---|---|---|---|---|---|
| Uncalibrated model | 0.794679 | 0.787876 | 0.0888445 | 0.5448 | 0.153413 |
| Platt scaling | 0.796196 | 0.794154 | 0.0384596 | 0.459308 | 0.146689 |
| Isotonic regression | **0.79724** | 0.796942 | **0.00774595** | **0.447345** | **0.142313** |
| Binning with PS | 0.793212 | **0.801376** | 0.00859279 | 0.449277 | 0.142469 |

Table 1: Platt scaling, isotonic regression, and binning with platt scaling improve calibration without sacrificing accuracy. See Appendix A.1 and A.2 for details on the calibration methods and the evaluation metrics.

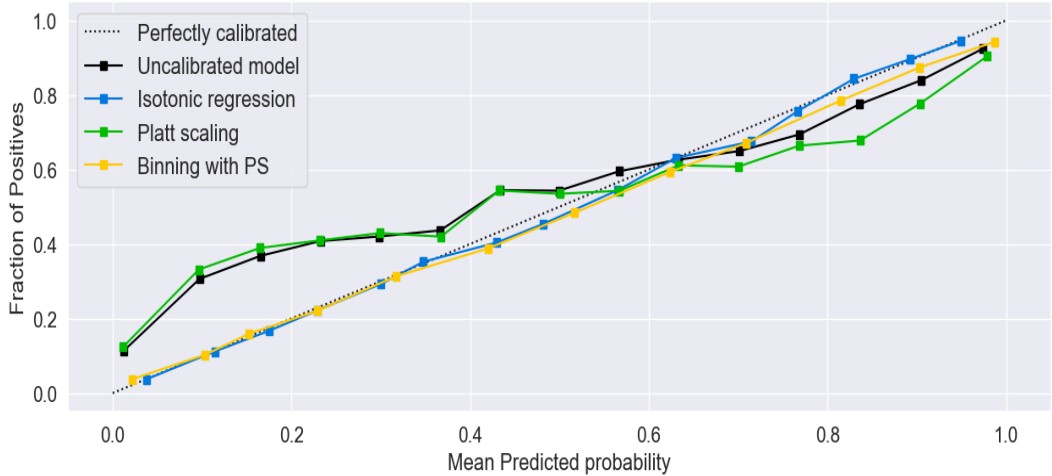

Figure 2: Reliability diagram of the uncalibrated and calibrated models. Any deviation from a perfect diagonal represents miscalibration. When the diagram is above the diagonal, the model is under-predicting the true probability, and if it is below, the model is over-predicting the true probability.

for constructing a model based on such balanced data. In total, this dataset contains 145,461 rows with 22 distinct features, detailing rain-related information. These features include elements such as date, geographical region, minimum and maximum temperatures, precipitation levels, wind speed and gust, hours of sunshine, etc. The target label for prediction is whether it will rain on the following day. We divided the data into 80% training, 10% validation, and 10% test.

We train a Multi-Layer Perceptron (MLP) neural network on the rain forecasting dataset. The architecture of the model composes four hidden layers with 64, 32, 16, and 8 neurons respectively. We used the rectified linear unit (ReLU) as the activation function (Zeiler et al., 2013). The optimizer used is *Adam*, which is a popular optimization algorithm for training neural networks (Kingma & Ba, 2015). The initial learning rate was set to 0.0001. The learning rate is adaptive, implying that it can dynamically adjust during training to enhance convergence. A batch size of 32 was set for each iteration of training. The training process spanned 2,000 iterations, ensuring sufficient time for convergence to be achieved.

We examined several popular methods for calibrating the trained model. Table 1 and Figure 2 exhibit the reliability diagram and a comparison of several evaluation methods between the models, as well as accuracy and balance (F1). Isotonic regression produced the most calibrated model without sacrificing the accuracy of the model. Therefore, we chose this method to calibrate the model in the evaluation survey we conducted with humans.

Overall, we tested 5 types of models in the survey:

1. Uncalibrated: the MLP model described above without any calibration method.
2. Calibrated version of the MLP model, with calibration done using isotonic regression.

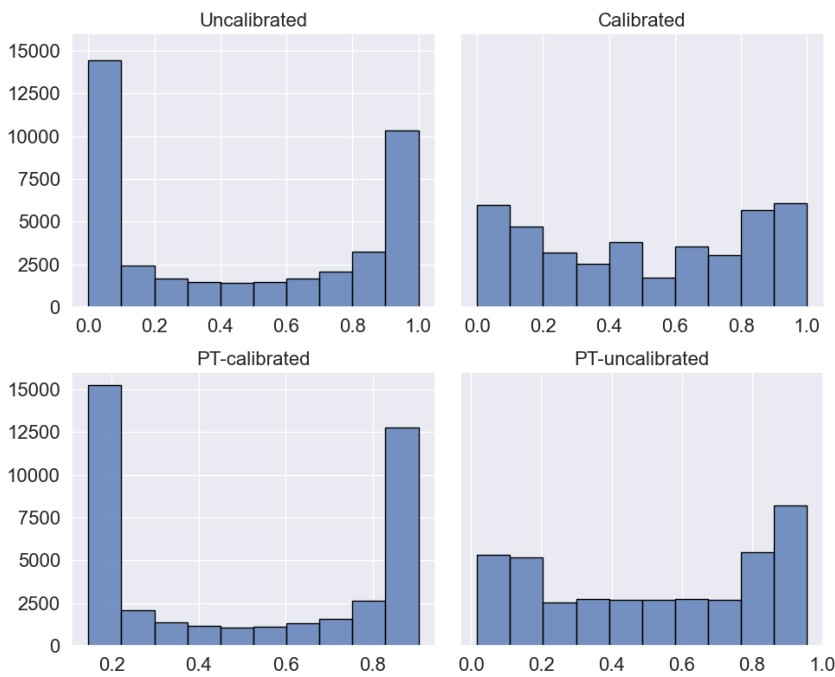

Figure 3: Probability distribution of models predictions on the validation set.

3. PT-calibrated: each calibrated probability is transformed using the prospect theory method as described in section 3. We used $\gamma = 0.71$, which is the fitting $\gamma$ value for USA residents according to a survey conducted by Rieger et al. (2017).

4. PT-uncalibrated: applying the prospect theory correction to the uncalibrated model.

5. Random PT-calibrated: similar to the PT-calibrated model except that the real outcome for each prediction is chosen randomly (rained or not). This model is added to verify that any differences between the PT-calibrated model and other models are not solely due to the reported values, but are in relation to the true predictions.

Figure 3 describes the distribution of predictions of all models on the validation set. The predictions were grouped into 10 bins. It can be seen that the uncalibrated model exhibits a tendency to generate predictions that approach 0 or 1. Conversely, the calibrated model aligns the predictions more faithfully with the true probabilities. When incorporating prospect theory into the calibrated model, it also yields predictions that are in proximity to 0 or 1, albeit to a lesser extent compared to the uncalibrated model. Recall that this is because humans interpret probabilities closer to 0 or 1 as being less extreme. So, we want to show them probabilities that are closer to 0 or 1 than the actual probabilities.

## 4.2 EXPERIMENTING WITH HUMANS

In the evaluation survey, we first explain to each participant that the goal of the survey is to test the trustworthiness of a rain forecasting system and that the system predicts the chance of rain on a particular day. We also ask the participants to assume that they are in a season and region where it rained on half of the days in past years.

Then, for 20 scenarios, each representing a different day, we present the participant 3 pages. On the first page, we present the chance that the rain forecast system predicts it will rain on that day. Participants are asked to assume they are planning an outdoor activity on that day and, based on the system's prediction, indicate how likely they are to cancel this activity. Participants can choose from five options on a Likert scale (Joshi et al., 2015), ranging from "not at all" (1) to "very much" (5). On the second page, the real outcome (whether it rained or not) is displayed along with an appropriate

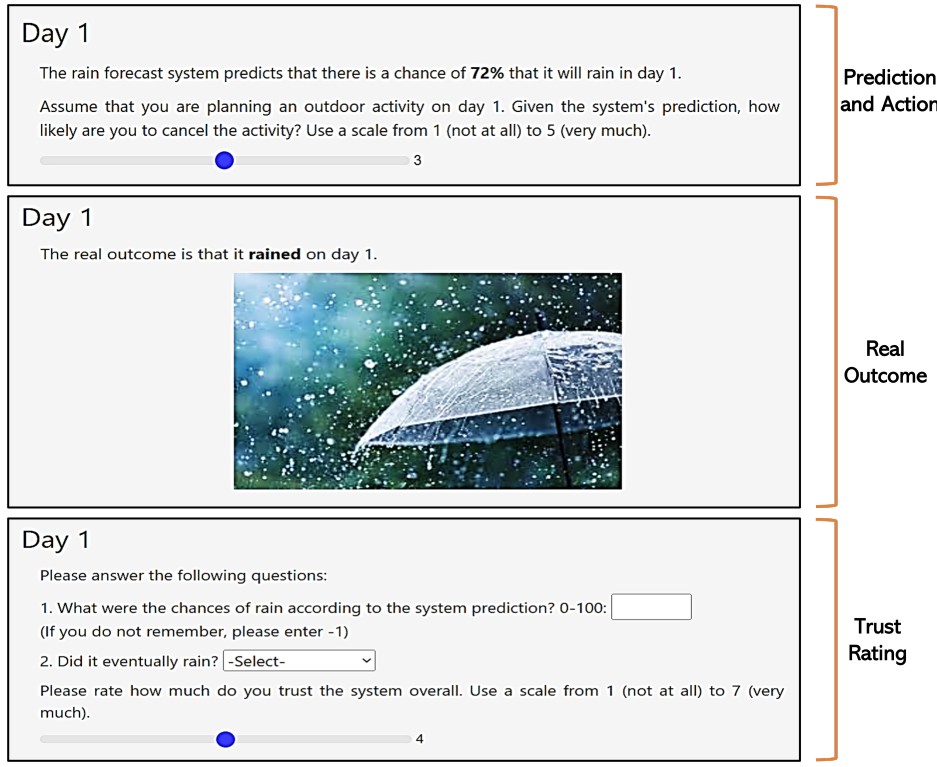

Figure 4: Interface of the experimental task in the survey.

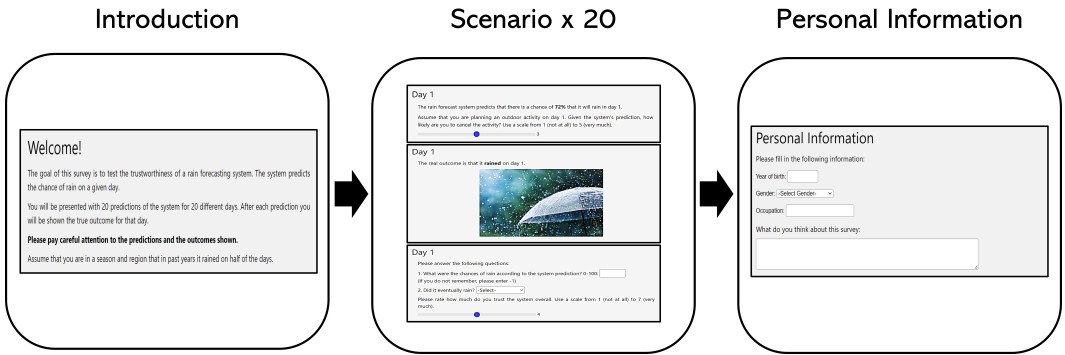

Figure 5: Flowchart of the survey.

visual image. On the third page, participants are asked to recall both the model prediction and the actual outcome. This step ensures that participants are actively engaged in considering the predictions and outcomes rather than simply advancing to the next page. Finally, participants are asked to rate their level of trust in the model. They can select from seven options on a Likert scale, spanning from "not at all" (1) to "very much" (7). Figure 4 is an example of these three pages for the first day.

We note that participants were asked to specify their degree of trust in the model's predictions and to decide whether they would cancel an event based on those predictions. This design was chosen to estimate participants' perceived reliability of the model. We hypothesize that while people may find it challenging to articulate their exact level of trust, observing the correlation between their actions and the model's predictions might offer a more accurate indication of their trust.

After presenting the scenarios, we asked each participant to provide personal information: age, gender, and occupation. Figure 5 depicts the flow of the entire survey for each participant.

Each of the five models was presented to 30 different participants, resulting in a total survey sample of 150 participants. Among the 150 participants, there are 86 males and 64 females, with an average age of 41 years. Each participant received a reward of $1. We set a requirement on Mechanical Turk that the location of the workers is the USA and their approval rate must be at least 99%. We did not require the Turkers to be masters.

This survey aims to analyze the perception of non-specialists in mathematics, computer science, statistics, or meteorology (laypeople). This demographic is of interest as their usage of machine learning is increasing; however, their lack of technical knowledge hinders their ability to accurately evaluate prediction quality. Previous research has indicated that laypeople may struggle with numerical data comprehension and may continue to rely on the recommendation of a machine learning model despite evidence suggesting low predictive performance (Peters et al., 2006; Reyna & Brainerd, 2008).

## 5 RESULTS

We assess the survey results using two key criteria: the average trust ratings of the models and the average correlation between the models' predictions and the participants' ratings of canceling the outdoor activity.

Figure 6a illustrates the average trust levels of participants across all scenarios for each model. As anticipated, the random model received the lowest trust rating. However, when comparing the other models, there is no notable difference in trust. In fact, except for the random model, there is no statistically significant difference between the models. Figure 6b compares the trust levels of the participants in the first scenario to their trust level in the last scenario for each model. While there seems to be some increase in the trust levels for some models and a decrease for others, these differences are not statistically significant. Overall, it cannot be concluded that either calibration or the addition of a prospect theory layer increases people's trust in the model.

We now examine the correlation between the model's predictions and the participants' likelihood of canceling the outdoor activity. To that end, we compute, for each participant, the correlation between the model's predictions and the participant's likelihood of canceling the outdoor activity, and we then average over all the correlations. As depicted by Figure 7, there are notable differences between the models. Specifically, the correlation observed for the PT-calibrated model is significantly higher than all other models ($p < 0.05$). However, as illustrated in the figure, calibration alone leads only to a slight improvement in correlation compared to the uncalibrated model. In addition, incorporating the prospect theory probability adjustments into the uncalibrated model only slightly enhances the correlation. Moreover, when the final results are random, and not related to the model predictions, the correlation is significantly lower (i.e., Random PT-calibrated is significantly lower than PT-

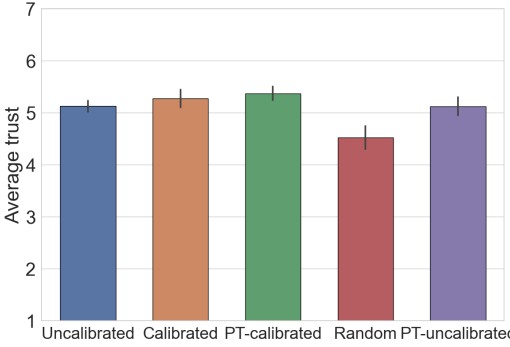 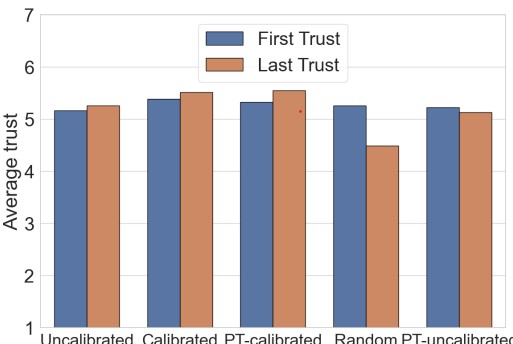

(a) Average of all trust ratings in all 20 scenarios. Error bars present the standard error.

(b) Average of trust ratings in the first and last scenarios only.

Figure 6: Average trust rating for each model.

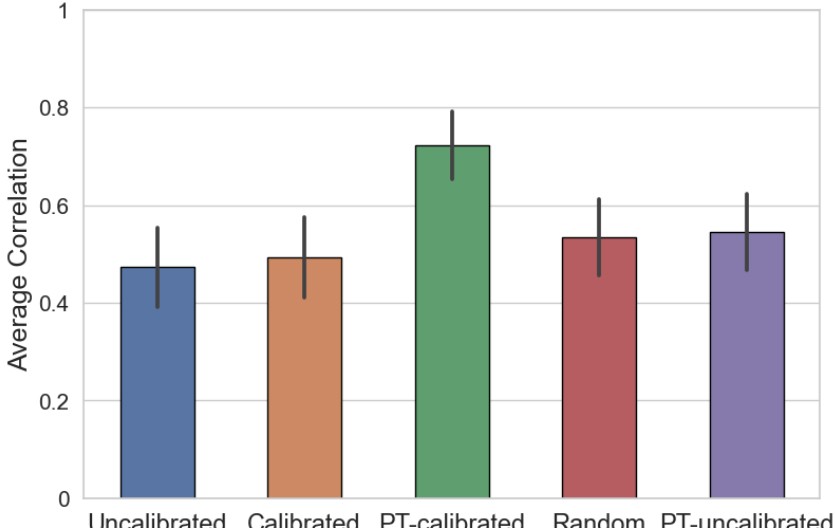

Figure 7: Average correlation between the model's predictions and the participants' rating to cancel the outdoor activity with Pearson correlation confidence interval (CI). The PT-calibrated model leads to a statistically significant increase in correlation compared to all other models, in particular, the calibrated model.

calibrated). We note that to check statistical significance we use a one-way ANOVA test between each two model results.

In other words, calibrating the model alone does not necessarily influence people to base their decisions on its predictions. However, adding to the calibrated model a layer that adjusts probabilities in line with people's expectations, as per prospect theory, encourages individuals to align their decisions with the model's predictions.

# 6 CONCLUSIONS AND FUTURE WORK

In this paper, we studied how people react to the probabilities predicted by a machine learning model. We considered if calibrating the probabilities increases the trust in the system, and whether it results in people taking actions that are better correlated with the system's prediction. Furthermore, we introduce a prospect theory-based correction that is applied to the model's calibrated probabilities. We show that while the trust in the system is not significantly affected by the method used, when asked to take action, the resulting correlation between the model's prediction and the human action is significantly higher for the model with calibration and prospect theory correction.

A limitation of our study is that we use a standard $\gamma$ value for the prospect theory correction. This $\gamma$ value is based on a domain-agnostic international survey. Estimating $\gamma$ precisely for the domain or population of interest (rain forecasting in Australia, for instance) is of natural interest. In future work, we intend to study if using domain-specific $\gamma$ further increases the size-of-effect in our findings.

Furthermore, our survey findings, as presented in this paper, indicate that simply calibrating models or incorporating a prospect theory layer does not have a significant impact on increasing people's trust in machine learning model predictions. Consequently, we recognize the need to explore alternative and innovative approaches focused on bolstering user trust. One promising avenue involves leveraging reinforcement learning techniques to train models explicitly designed to enhance user trust in the predictions they provide. This exploration could open up exciting possibilities for improving the acceptance and reliability of machine learning models in practical applications.

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

# A APPENDIX

## A.1 CALIBRATION METHODS

The following are common methods to calibrate a model. All methods are post-processing steps that produce calibrated probabilities. That is, given a model's output $p_i$ on sample $x_i$, they produce a calibrated probability $q_i$. To avoid unwanted bias, each method requires a validation set, which can also be used for hyperparameter tuning.

1. **Platt scaling** (Platt, 1999). A parametric approach that assumes a sigmoidal relationship between the model's outputs and the true probabilities. It trains a logistic regression model on the validation set, where the original model's outputs are used as features. That is, it learns scalars $a, b \in \mathbb{R}$ and outputs $q_i = \sigma(ap_i + b)$ as the calibrated probability. Platt scaling is particularly effective for max-margin methods such as SVMs and boosted trees, which show sigmoidal distortions in their predicted probabilities (Niculescu-Mizil & Caruana, 2005).

   Platt scaling can be extended to multiclass models by applying to the model's logits vector, $z_i$, a linear transformation $Az_i + b$, where $A \in \mathbb{R}^{K \times K}$ and $b \in \mathbb{R}$. The calibrated probability of the predicted class is $q_i = \max_k \sigma \left(Az_i + b\right)^{(k)}$ (Guo et al., 2017).

2. **Binning** (Zadrozny & Elkan, 2001). A simple non-parametric approach. It sorts all the model's predictions and divides them into $M$ interval bins. The bin boundaries are either chosen to be equal length intervals or to equalize the number of samples in each bin. Given a prediction $p_i$, the method finds the bin containing that prediction and returns as output the fraction of positive outcomes in the bin. This method has several limitations, including the need to define the number of bins and the fact that the bins and their associated boundaries remain fixed on all predictions. (Naeini et al., 2015) proposed BBQ, a refinement of the binning method using Bayesian model averaging.

3. **Isotonic regression** (Zadrozny & Elkan, 2002). A more general non-parametric approach, that only assumes a monotonic increasing relationship between the model's outputs and the true probabilities. It utilizes isotonic regression (Robertson et al., 1988) to learn an isotonic (non-decreasing) function $f$ to map $q_i = f(p_i)$. A common algorithm used for computing the isotonic regression model is the PAV algorithm (Ayer et al., 1955). Isotonic regression is a more powerful calibration method that can correct any monotonic distortion, however, is more prone to overfitting (Niculescu-Mizil & Caruana, 2005).

   Isotonic regression can be extended to multiclass models, by reducing the multiclass problem into a set of binary problems. A well-known approach to this end is one-against-all, in which a classifier is trained for each class using as positives the examples that belong to that class, and as negatives all other examples. At test time, we obtain probability vector $[q_i^{(1)}, ..., q_i^{(K)}]$ where $q_i^{(k)}$ is the calibrated probability for class $k$. The final prediction probability $q_i$ is the max of the vector normalized by $\sum_{k=1}^{K} q_i^{(k)}$. Another common approach is all-pairs (Allwein et al., 2000).

4. **Temperature scaling** (Guo et al., 2017). A simple extension of Platt scaling that learns a single Temperature parameter $t > 0$ to rescale the logit vector $z_i$. The calibrated probability of the predicted class is $q_i = \max_k \sigma \left(z_i/t\right)^{(k)}$. Temperature scaling does not affect the model's accuracy as $t$ does not change the maximum of the softmax function.

5. **Ante-hoc methods**. An alternative to post-hoc calibration is to modify the classifier learning algorithm itself. MMCE (Kumar et al., 2018) trains neural networks by optimizing the combination of log loss with a kernel-based measure of calibration loss. SWAG (Maddox et al., 2019) models the posterior distribution over the weights of the neural network and then samples from this distribution to perform Bayesian model averaging. (Milios et al., 2018) proposed a method to transform the classification task into regression and to learn a Gaussian Process model.

## A.2 EVALUATING CALIBRATION

To evaluate the calibration of the model from a finite set of $n$ samples, the predicted probabilities are grouped into $M$ interval bins of equal size. Let $B_m$ be the set of samples whose predicted

probability falls into bin $m$. The accuracy of $B_m$ is

$$acc(B_m) = \frac{1}{|B_m|} \sum_{i \in B_m} 1(\hat{y}_i = y_i),$$

where $\hat{y}_i$ and $y_i$ are the predicted and true class labels for sample $i$. The average confidence of $B_m$ is

$$conf(B_m) = \frac{1}{|B_m|} \sum_{i \in B_m} \hat{p}_i,$$

where $\hat{p}_i$ is the confidence score the model assigns to sample $i$ for belonging to class $y_i$. A perfectly calibrated model will have $acc(B_m) = conf(B_m)$ for all $m \in [M]$.

The following are common methods to evaluate the calibration of a model.

1. **Reliability Diagram** (Murphy & Winkler, 1977): is a visual representation of model calibration. These diagrams plot for every bin $m \in [M]$ the accuracy of $B_m$ as a function of the confidence of $B_m$. If the model is perfectly calibrated, then the diagram should plot the identity function. Any deviation from a perfect diagonal represents miscalibration. When the diagram is above the diagonal, the model is under-predicting the true probability, and if it is below, the model is over-predicting the true probability. See figure 2 for an example of a reliability diagram.

2. **Expected Calibration Error (ECE)** (Naeini et al., 2015). While reliability diagrams are useful visual tools, it is more convenient to have a scalar summary statistic of calibration. ECE approximates the difference in expectation between confidence and accuracy by taking a weighted average of the bins' accuracy/confidence difference.

$$ECE = \sum_{m=1}^{M} \frac{|B_m|}{n} \left| acc(B_m) - conf(B_m) \right|$$

A well-calibrated model will have a small ECE, indicating that its predicted probabilities closely reflect its actual performance, whereas a poorly calibrated model will have a large ECE.

3. **Maximum Calibration Error (MCE)** (Naeini et al., 2015). In high-risk applications, confident but wrong predictions can be especially harmful. In such cases, we may wish to minimize the worst-case deviation between confidence and accuracy. MCE estimates an upper bound of this deviation.

$$MCE = max_{1 \in [M]} \left| acc(B_m) - conf(B_m) \right|$$

4. **Overconfidence Error (OE)**. Another method that measures the overconfidence of a model. This metric penalizes predictions by the weight of the confidence, but only when confidence exceeds accuracy, and thus overconfident bins incur a high penalty.

$$OE = \sum_{m=1}^{M} \frac{|B_m|}{n} \left[ conf(B_m) \cdot max\left( conf(B_m) - acc(B_m), 0 \right) \right]$$

5. **Negative log-likelihood (NLL)**. Measures the difference between the confidence score and the true probabilities of the outcomes, as expressed in the likelihood function.

$$NLL = -\sum_{i=1}^{n} \log(\hat{p}_i)$$

The lower the NLL, the better the calibration of the model. Note that the negative log-likelihood is also equivalent to the cross-entropy loss, which is commonly used in machine learning for classification tasks (De Boer et al., 2005).

6. **Brier Score (BS)** (Brier, 1950). is the mean squared error between the confidence scores and the true class labels.

$$BS = \frac{1}{n} \sum_{i=1}^{n} \sum_{k=1}^{K} (y_i^{(k)} - p_i^{(k)})^2,$$

where $K$ is the number of classes, $y_i^{(k)}$ is a binary variable that indicates whether the sample $x_i$ belongs to class $k$ and $p_i^{(k)}$ is the probability the model assigns to sample $x_i$ for belonging to class $k$. The Brier score ranges from 0 to 1, where 0 indicates perfect calibration and 1 indicates the worst possible calibration.

