# OpenReview forum: "Does Calibration Affect Human Actions?"
_ICLR.cc/2024/Conference — Submitted to ICLR 2024_

### Official Review · Reviewer_NQcN · 2023-11-04

**Soundness:** 2 fair
**Presentation:** 3 good
**Contribution:** 3 good
**Rating:** 3
**Confidence:** 3

**Summary:**

This paper explores the link between calibration and human trust in an AI model.
The authors state that according to prospect theory, humans consume probabilities based on a reference point to their current situation.
Based on this the authors develop a calibration approach (PT-calibrated) that re-weighs probabilities based on how humans subjectively consume them.
To evaluate their approach they develop a human survey where participants are, among other aspects, queried about their trust in the model based on confidence.
The authors show that the model whose calibrated confidences were further treated with the weighting function (based on prospect theory) is best trusted by the participants and its predictions best correlate with human decisions.
The authors also find the largest increase in human trust between the first and last questions of the survey by PT-calibrated, however these results are not statistically significant.

**Strengths:**

- The paper examines a very important problem: the link between confidence calibration and how humans make judgments using these confidence scores
- The paper shows how a reweighting function (with ideas from decision theory) that can reweight confidences elicits more trust from humans than a simple calibrated model
- The paper's ideas and results are crucial to creating trustable ML systems and would be very interesting to these communities

**Weaknesses:**

- I think the paper's experimentation is lacking.
    - The current experimental setup is much too simplistic: 1. Just asking the users how much they trust the system can result in a lot of noise especially as users have no reason to be faithful. It seems that prior works usually measure some proxy for trust [1], or simulate an environment where where participant's trust is linked to some monetary risk/reward [2.3]
    - The authors show experiments on a single task, also the authors ignore the temporal effects of changing trust as the participant interacts with the system.
- A lot of experimental design choices (eg Likert scale to quantify trust) seem to differ from prior works that examine human trust. Perhaps the authors could spend more time justifying them

Minor:
- Table 1, why not round down to some number of significant digits?

Overall, I think the paper has some very interesting ideas but is still not mature enough for acceptance owing to the lack of thorough experimentation.

[1] Zhang, Yunfeng, Q. Vera Liao, and Rachel KE Bellamy. "Effect of confidence and explanation on accuracy and trust calibration in AI-assisted decision making." Proceedings of the 2020 conference on fairness, accountability, and transparency. 2020.
[2] Vodrahalli, Kailas, Tobias Gerstenberg, and James Y. Zou. "Uncalibrated models can improve human-ai collaboration." Advances in Neural Information Processing Systems 35 (2022): 4004-4016.
[3] Gonzalez, Ana Valeria, et al. "Human evaluation of spoken vs. visual explanations for open-domain qa." arXiv preprint arXiv:2012.15075 (2020).

**Questions:**

- How much time did each participant take to complete the survey on average?
- How do the authors ensure that the participant responses were faithful?
- Did the participants receive $1 per question or for 30 questions?

---

> ### Author Response · Authors · 2023-11-19
>
> - We agree with the reviewer that the self reported trust by the participants is noisy. Indeed, our results clearly support this claim. Therefore, our results are based on the participants' report on how likely they are to cancel their outdoor activity based on the model’s prediction. Recall that we show that there is a statistical significant difference between the different treatment groups, thus, it is quite unlikely that these differences are due only to noise.
> - We plan to add results for another task.
> - We used different Likert scales during the experiment (1-5, and 1-7) to ensure that the participants think and answer each question separately (so that the answer to one question will have a lesser effect on the other question).
>
>
> Answers:
> 1. The average work time of the participants is 9.5 minutes.
> 2. Recall that we asked the participants to report the values that the model predicted and the actual outcome. This encourages the participants to answer seriously; we will emphasize this in the paper. Furthermore, all participants in the survey were required to have at least 99% approval rate (on other tasks on Mechanical Turk).
> 3. $1 for all questions, but note that there were 20 questions, not 30 questions.

---

### Official Review · Reviewer_Yu3U · 2023-11-05

**Soundness:** 2 fair
**Presentation:** 2 fair
**Contribution:** 2 fair
**Rating:** 5
**Confidence:** 4

**Summary:**

The paper examines the effectiveness of the probabilistic calibration as to how it affects the human decision making. In particular, the paper studies if the humans (decision makers) are willing to change their decision making depending on how (and what kind of) forecast is revealed to them for the relevant event. Along with standard calibrated and uncalibrated forecasts, the paper also employs post-hoc corrections to the forecasts based on the prospect theory in behavioural economics. Overall, the paper finds that there are no significant differences in the reported users' trust for different forecasts, but forecasts involving prospect theory correction shows better correlation to users decisions.

**Strengths:**

1. The paper asks a relevant question. Traditional calibration is usually considered as the de-facto measure of reliability in popular machine learning literature. However, machine learning prediction systems are not built in isolation and have major implications how they affect human decision systems. Thus, studying the usability of calibration to actual human subjects is an insightful research question.
2. The introduction of prospect theory based post-hoc correction is also interesting to make the forecasts better aligned to human interpretations.

**Weaknesses:**

1. One of the crucial limitations of the paper is lack of thorough description of human study conducted. The paper claims that "there is no reported difference in the level of trust reported by the participants". However, without further information on the nature of instructions / guidelines provided to the human subjects, it could very well be the case that the subjects of this study behaved randomly (which is not an uncommon phenomenon, and is usually controlled for in user studies by designing good incentive mechanisms). The paper (in the current form) does not delve much deeper whether the measures were taken to control random behaviour.  My opinion is also informed by correlation in Figure 7, where the difference between random and calibrated / uncalibrated is not that different.

Overall, I think the paper is interesting. However, due to the above concern, I'm hesitant to fully rely on the user study.  I'm happy to hear more from the authors.

**Questions:**

1. The paper misses some of the relevant literature on the implications of calibration to decision making [1,2].



[1] Benz et al. Human-Aligned Calibration for AI-Assisted Decision Making.
[2] Rothblum et al. Decision-Making under Miscalibration.

---

> ### Author Response · Authors · 2023-11-19
>
> - The instructions to the participants appear at the beginning of section 4.2 and in Figure 5. However, we notice that the text in the figure is hard to read; this will be corrected.
>
> - Recall that we asked the participants to report the values that the model predicted and the actual outcome. This encourages the participants to answer seriously; we will emphasize this in the paper. Furthermore, all participants in the survey were required to have at least 99% approval rate (on other tasks on Mechanical Turk).
>
> - The random model is not arbitrary in its predictions. It provides probabilities like the PT-calibrated model, with the distinction that the determination of whether it rained on a specific day was randomly generated. As a consequence, the probabilities it produces tend to be extreme, i.e., either close to 0 or 1. We believe that it is more convenient for people to listen to a model that provides predictions at the extremes, as this diminishes uncertainty. However, when the level of trust also take into account, the "random" model falls significantly short compared to other models. We intend to change the name of the model since "random" does not accurately reflect how it works.
>
> Answers:
> We thank the reviewer for the pointers and will add them to the paper.

---

### Official Review · Reviewer_7xFz · 2023-11-10

**Soundness:** 3 good
**Presentation:** 3 good
**Contribution:** 2 fair
**Rating:** 6
**Confidence:** 4

**Summary:**

The paper proposes correcting calibrated confidence scores based on Kahneman and Tversky's prospect theory as to adjust confidence scores in line with how people perceive probabilities. For example, reporting a 80% confidence score as 90%, as per prospect theory, a 90% probability would be perceived as 80%. In a study with human participants, they compare the impact of their proposed approach against a calibrated model and 3 other baselines. While there is no significant difference in terms of reported trust between the models, the correlation between decisions and predictions increases for their approach compared to the baselines.

**Strengths:**

- The proposed idea of using prospect theory on top of calibration to help align human perception with the model's predictions is nice and seems novel.
- The experimental results support the claim of the paper that using prospect theory together with calibration increases correlation of individuals decisions with the model's prediction.

**Weaknesses:**

- The methodological contribution itself is relatively small, the application of prospect theory to the problem is quite straightforward.
- The study setting is somewhat limited in that the participants have to make decisions based on the predictions of the model only and have no other information available. This doesn't seem to be realistic in most assisted decision making scenarios, where the individual could ignore the model if they do not trust it and base the decision on their own knowledge (e.g., the tasks in Vodrahalli et al. 2022). It would be interesting to know if we can expect that calibration+prospect theory to also lead to higher correlation in such tasks where the individual has the same (or other/additional) information available as the model.
- Some parts of the study design and the evaluation were unclear to me (see questions).

**Questions:**

- For some parts of the evaluation it is unclear which data was used: Are the results of Table1 and Figure 2 from the data in the validation set? Was the test set used in the survey with the human participants?
- It would be nice if the authors could point out earlier that the $\gamma$ value chosen is not specific for this task. This was unclear when described in page 6 and only discussed much later in the conclusion.
- It is interesting that, even though individuals reported to trust the random model less, the correlation of the random model's prediction with the individuals' decisions is higher than the calibrated and uncalibrated model's correlation (Figure 6a and 7). Do the authors have an intuition why this is the case?

---

> ### Author Response · Authors · 2023-11-19
>
> In the current version of the paper, we attempted to use the “cleanest” environment for testing our hypothesis. However, we are planning to run additional experiments in a new domain, in which we will allow the participants to gather additional information.
>
> Answers:
> 1. Table 1 and Figure 2 were both calculated on the validation set. In the current manuscript, the test set results are not reported. We will add them to the next version.
> 2. We will point this out earlier, as the reviewer suggested.
> 3. The random model is not arbitrary in its predictions. It provides probabilities like the PT-calibrated model, with the distinction that the determination of whether it rained on a specific day was randomly generated. Consequently, the probabilities it produces tend to be extreme, either close to 0 or 1. We believe that it is more convenient for people to listen to a model that delivers predictions at the extremes, as this diminishes uncertainty. We intend to change the name of the model since "Random" does not accurately reflect how it works.

---

> > ### Comment · Reviewer_7xFz · 2023-11-20
> > **Additional Questions and Comments**
> >
> > I thank the authors for the clarifications. I have two related questions/comments.
> >
> > - **Related to answer 1:** It is still unclear to me if the 20 scenarios in the experiment were taken from the whole data set or from the test set. Also I understood that the same 20 scenarios were shown to all participants, is this correct?
> >
> > - **Related to answer 3:**
> > From the reported Figures 3 and 7, it seems that having predictions close to 0 or 1 is not directly causing a higher correlation of predictions and individuals decisions since the uncalibrated model has predictions close to 0/1 but lower correlation than the PT-uncalibrated model which has less extreme predictions (and both models have the same level of reported trust). I think adding additional experiments in a new domain might provide more insights into these results. For example, I expect that if the human has additional information to make a decision, the correlation of the "Random" model will decrease along with the decrease in reported trust, because the human can rely on their own knowledge to make make the decision and not only the AI prediction.
> > Do the authors already have some preliminary results of the additional experiments?
> >
> > Overall, I think the current experiment setup is a good start for the evaluation but it is missing some key components (such as the expertise of the human) to better understand the effects of prospect theory and calibration on decision making.

---

### Meta-Review · Area_Chair_gS2v · 2023-12-06

**Metareview:**

The paper proposes using prospect theory to correct calibrated confidence scores in a human-ai collaboration scenario. In addition, they run a human subject study to show that, by using this correction, the correlation between human decisions and the model's prediction increases. All the reviewers indicated that the work has merit and a lot of potential. However, all of them highlighted that several points for improvement, most prominently regarding the human subject studies. The rebuttal/discussion did not completely cleared out the reviewers' concerns. I encourage the authors to revise their manuscript in light of the reviewers' comments and resubmit their work to another top tier conference.

**Justification For Why Not Higher Score:**

There are significant points for improvement highlighted by the reviewers

**Justification For Why Not Lower Score:**

N/A

---

### Decision · Program_Chairs · 2024-01-16

Reject